# Using Ensembles for Accurate Modelling of Manufacturing Processes in an IoT Data-Acquisition Solution

**José Luis Garrido-Labrador** [1] , **Daniel Puente-Gabarri** [2] , **José Miguel Ramírez-Sanz** [1] , **David Ayala-Dulanto** [2] **and Jesus Maudes** [1,*]

1   Departamento de Ingeniería Informática, University of Burgos, 09001 Burgos, Spain;
    jlgarrido@ubu.es (J.L.G.-L.); jmrsanz@ubu.es (J.M.R.-S.)
2   Departamento de Ingeniería Informática, Fundación Centro Tecnológico de Miranda de Ebro,
    09200 Miranda de Ebro, Spain; danielpuente@ctme.es (D.P.-G.); dayala@ctme.es (D.A.-D.)
*   Correspondence: jmaudes@ubu.es

**Abstract:** The development of complex real-time platforms for the Internet of Things (IoT) opens up a promising future for the diagnosis and the optimization of machining processes. Many issues have still to be solved before IoT platforms can be profitable for small workshops with very flexible workloads and workflows. The main obstacles refer to sensor implementation, IoT architecture, and data processing, and analysis. In this research, the use of different machine-learning techniques is proposed, for the extraction of different information from an IoT platform connected to a machining center, working under real industrial conditions in a workshop. The aim is to evaluate which algorithmic technique might be the best to build accurate prediction models for one of the main demands of workshops: the optimization of machining processes. This evaluation, completed under real industrial conditions, includes very limited information on the machining workload of the machining center and unbalanced datasets. The strategy is validated for the classification of the state of a machining center, its working mode, and the prediction of the thermal evolution of the main machine-tool motors: the axis motors and the milling head motor. The results show the superiority of the ensembles for both classification problems under analysis and all four regression problems. In particular, Rotation Forest-based ensembles turned out to have the best performance in the experiments for all the metrics under study. The models are accurate enough to provide useful conclusions applicable to current industrial practice, such as improvements in machine programming to avoid cutting conditions that might greatly reduce tool lifetime and damage machine components.

**Keywords:** ensembles; unbalanced datasets; internet of things; rotation forests; milling

## 1. Introduction

Over the past 10 years, different technologies have boosted data-acquisition, communications, and processing capabilities. This strong development has led to the inauguration of a new concept: the Internet of Things (IoT). Although this concept might have direct applications in the daily life of the public, its successful implementation in industrial environments seems to be a more complex issue, as recent reviews have outlined [1]. This is the case of machining workshops, where many factors limit the range of IoT solutions. First, the integration of new sensors in existing machines is not easy, as durable machine-tools are usually designed for a long life and most existing machines were built before the development of the IoT or the Industry 4.0 paradigms [2]. Therefore, communication capabilities and integrated sensors built into machine-tools are very limited. In those cases, the only

way to extract information from them would be through the machine's CNC (Computer Numerical Control). However, the CNC is not often available, given that its primary function is to control the machine-tool. Therefore, the most common solution is to access the PLC (Programmable Logic Controller) of the machine. Reading the PLC parameters is one way of extracting many parametric values from the CNC. This solution has been widely used during the last decade; i.e., for developing adaptive remote controllers for milling machines through an internet connection [3].

The first element for suitable IoT solutions in small workshops should therefore be a Data Acquisition Platform (DAP) connected to the PLC of the machine. The first DAP for manufacturing tasks was demonstrated over 20 years ago [4] for tool condition monitoring. However, the opportunity of setting up a real implementation was never demonstrated, due mainly to the very limited access to commercial CNCs at that time. To overcome this limitation, open architecture CNCs, once very rare in industrial workshops, were used in most studies over the past twenty years. In very recent years, some studies have described the new communication capabilities incorporated in commercial CNCs.

Having established reliable solutions for data communication, the research focus moved on towards the definition of the best Key Performance Indicators extracted from the IoT platforms for manufacturing optimization.

However, the data-acquisition stage is, however, not the only challenge for machine-workshop IoT solutions. Data processing and analysis are also subject to very restricted conditions and data features. As will be explained, the analysis of workshop machining processes will often have to contend with incomplete and unbalanced datasets. Besides, the data will have too many inputs, lessening its reliability. It will therefore be necessary to reduce the number of inputs and to eliminate repeated instances without losing information.

Machine-learning techniques have many capabilities that are especially suitable to overcome these limitations. First, they generalize models to new conditions, thereby reducing the number of expensive experimental tests to be performed. Second, machine-learning techniques can extract useful information for unbalanced datasets. Third, machine-learning techniques reduce the number of features without losing information. Fourth, machine-learning techniques are able to complete missing attributes, due to sensor malfunctions and data-transmission errors.

Nevertheless, the studies on machine-learning algorithm applications to predict machining-process performance have been demonstrated in laboratory datasets. Datasets generated under laboratory conditions have some extremely different features to those generated in real workshops. Under laboratory conditions, a very small number of inputs are varied from one experiment to the next, there is almost no experimental repetition, the experimental conditions are carefully selected, mainly by factorial or Taguchi experimental design, and all inputs and outputs are carefully measured, and validated before the next experiment is performed, as outlined by the most exemplary reviews [5,6]. However, as Bustillo et al. demonstrated [7], under industrial conditions, most of the experiments refer to the same cutting conditions, a very broad range of parameters are at the same time varied, and the data present many empty values and values of limited confidence.

Most of these features can be joined in one dataset property: artificially balanced datasets. Balance means that there is a similar proportion of all the tested conditions and the values of the outputs in the dataset. This condition is however far from the industrial reality, where most of the instances in the acquired dataset will show a normal behavior and very few will show an abnormal behavior, independent of the considered machining process or output that is to be predicted.

The starting point of this study is completely unlike previous works: the aim is to design a reliable data-acquisition system and to connect it to a machining center in a workshop, acquiring data almost under blind conditions. The machine will be monitored over a sufficient period of time to gain a general overview of its operation, in this case 3 months. The machine has a commercial, non-open architecture CNC that will ensure a suitable IoT solution for all other machine-tools. Then, different machine-learning algorithms will be used to model continuous features, such as motor temperatures, and some discretized features, such as machine state and working mode.

The novelty of this approach is not that it develops a breakthrough approach for data acquisition and IoT implementation, nor is it in the design of new machine-learning algorithms. It refers to both the design and the validation of a reliable IoT solution for its immediate implementation in real workshops. The solution is based on the most suitable machine-learning algorithms for each of the proposed industrial tasks (identification of temperature patterns and machine working states). This suitability will be proven with a real and extensive dataset, where different machining processes are mixed with no clear identification: milling, drilling, etc. Our solution, as previously outlined, is different to the previously presented proposals, because it is not built up from laboratory datasets and specifically designed machining tests (i.e., only face-milling tests at different cutting speeds), but from real datasets with many repetitions, different cutting conditions and processes, etc. Besides, no pre-processing techniques are applied to the dataset, to assure that the selected machine-learning algorithms will be ready to process real information later on without the intervention of a human expert.

The existing bibliography, summarized in Section 2, proposes solutions for certain industrial problems (e.g., tool breakage or wear monitoring in drilling, milling or turning, surface quality or dimensional accuracy prediction in machined workpieces, etc.). However, in this case, the solution is open to extract any useful information, merely by changing the inputs and outputs of the trained model and without any previous classification of the cutting process that took place. This feature is assured through continuous and discretized outputs of industrial interest. Therefore, the novelty of this research, rather than a solution for a specific manufacturing task, proves a more generic solution and a suitable approach to extract useful information from real manufacturing data.

The paper will be organized as follows. A brief state of the art of IoT platforms for machining workshops and some examples of the most common machine-learning algorithms used on those platforms will be included in Section 2. In Section 3, the IoT data-acquisition platform and the machine connected to it will be presented. Then, in Section 4, the dataset extracted from the data-acquisition platform will be presented with the machine-learning techniques to model the two datasets. Special attention will be paid to the most suitable metrics to evaluate model performance. The results of such modeling and the industrial use of the best model will be presented in Section 5. Finally, the most relevant results will be summarized in the conclusions (Section 6) and pointers will be given for future lines of research.

## 2. State of the Art

### 2.1. Data Acquisition Platforms in IoT Solutions

After the first attempt to build a DAP applied to the manufacturing tasks by Ebrahimi [4], the first solutions based on open architecture CNCs were developed. Ferraz et al. [8] proposed the connection of an open architecture CNC to a lathe machine with an intelligent system, to overcome the tool-wear effect in workpiece quality. To do so, they installed a linear variable differential transformer and acoustic emission sensors on the lathe machine, demonstrating that on-line workpiece quality monitoring and control can be performed with Ethernet and Internet connections. Along the same lines, Frumusanu et al. [9] presented a stability control system for in-process modification, to avoid chatter vibrations and to achieve the highest process performance in turning operations on an open CNC transversal lathe. By doing so, they were able to maintain performance and to avoid chatter vibrations for a fixed turning process.

As presented in Section 1, commercial CNCs have only recently included communication capabilities that have been studied in very recent works. Mourtzis et al. [10], proposed an approach for machine-tool energy consumption estimation, based on real-time monitoring measurements using wireless sensor networks. This solution overcomes the lack of data acquisition and transmission in machine-tools previously installed in real workshops. Within the same research area, Zhong et al. [11] developed a Radio Frequency Identification framework to overcome communication limitations in mass-production workshops, thereby assuring component traceability. Finally, Schafer et al. [1]

reviewed the most recent publications in IoT solutions for workshops, concluding that most of the existing solutions have limited capabilities and real-time monitoring, due to the low sensorized rate of existing machine-tools.

Now that reliable solutions for data communication have been developed, the research emphasis has shifted towards defining the best key performance indicators extracted from IoT platforms for optimal manufacturing [12]. Chen et al. [13] proposed a combination of machine state, energy consumption, and cutting tool identifier. Their solution reduced energy consumption and energy cost by effectively managing the defined machining process, though it is very dependent on reliable identification and supervision of cutting processes. In a more advanced work, Chen et al. [14] tested this general strategy in a real workshops with 25 machine tools, 24 LED lamps, 10 ventilation fans, 5 air conditioners, 2 transport devices, and 2 air compressors. They demonstrated that energy indicators can be improved by considering efficiency indicators, although once again, these indicators require an extensive knowledge of the cutting processes under execution.

## 2.2. Machine Learning Techniques Applied to Machining Optimization

The following works explore the suitability of machine-learning techniques in depth, to alleviate the limitations of machining-datasets:

- Regarding the reduction of the number of expensive experimental tests, Oleaga et al. [15] demonstrated that some machine-learning algorithms, for example, Random Forest ensembles, can provide accurate prediction models for critical depth of cut-and-chatter frequency in milling operations with a smaller number of experimental tests than experimental or analytical models.

- With respect to the extraction of useful information from imbalanced datasets, Bustillo and Rodriguez [16] demonstrated that some of those techniques, such as ensembles, can overcome unbalanced data in an extensive experimental dataset. They validated this approach to unbalanced data through industrial breakage detection of multitooth tools in real industrial datasets, showing successful detection of 59 insert breakages from a total dataset of 30,000 mechanized crankshafts.

- As for the reduction of the number of features without losing information, Grzenda et al. [17] demonstrated that Multilayer Perceptrons can make reliable predictions of surface roughness for face-milling operations, following dataset dimensionality reduction. They reduced the number of accelerometers needed in this case for reliable machine process monitoring. Grzenda and Bustillo [18] proposed the use of a genetic algorithm with neural networks to identify the best set of inputs to provide accurate prediction models for surface quality in high-torque face milling operations, reducing the data-acquisition costs within industrial environments.

- The capability of machine-learning techniques to complete missing attributes, due to sensor malfunction or data-transmission error has also been studied in a few works. Grzenda et al. [19] demonstrated that Genetic Algorithms and Multilayer Perceptrons can complete damaged datasets in deep drilling operations of steel components to predict borehole roughness. Besides, machine-learning techniques have proved their capability to create especially designed visual models for real-time visual processing of many manufacturing processes. Teixidor et al. [20] used different machine-learning algorithms, among which k-Nearest Neighbors, neural networks and decision trees, to model some outputs of industrial interest in pulsed laser micromachining of micro geometries, such as dimensional accuracy, surface roughness and material removal rate. They demonstrated reliable models of immediate industrial application by means of decision trees, which can process direct rules or 3D-Charts that optimize process parameters.

- The suitability of machine-learning techniques to build reliable prediction models for different machining-process cutting outcomes has been widely demonstrated. Bustillo et al. [21] proposed the use of Bayesian Networks for breakage detection of cutting tools in crankshafts machining. They showed the flexibility of these types of networks to extract process information of direct

industrial use by means of inquiries forwarded to the Bayesian Network when evaluating tool wear in a discretized way: broken or normal. Karandikar et al. [22] proposed a Bayesian inference method to evaluate tool life for end milling of AISI 1018 in a continuous way, instead of with a discretized number of levels. Along the same lines, but using Multilayer Perceptrons, Mikołajczyk et al. [23] described the use of MLP-based automatic image analysis for assessing tool wear. Their results promised a good correlation between the new methods and the commonly used optically measured VB index for the entire life range of the tools.

The same two strategies (continuous or discretized output) have been proposed for the two main workpiece-related quality indicators: surface roughness and dimensional accuracy. Grzenda et al. [17] built accurate Multilayer-Perceptron models for surface roughness prediction in cast-iron face-milling operations. Facing the same task, Rodríguez et al. [24] proposed surface roughness prediction in face milling, through the use of decision trees for their immediate implementation by process engineers in workshops, instead of black box models such as Multilayer Perceptrons (MLPs). Bustillo et al. [25] defended the advantages of Bayesian Networks to predict surface roughness in deep drilling operations with steel components, in this case, by discretizing surface roughness using an industrial standard: the ISO 4288:1996.

Workpiece dimensional accuracy as a continuous output has been also modeled with neural networks and decision trees in pulsed laser micromachining of Hardened AISI H13 tool steel by Teixidor et al. [20]. In contrast, Ferreiro et al. [26] demonstrated that machine-learning algorithms, especially Bayesian Networks and Decision trees, were more accurate than mathematical models for the detection of burr during high-speed drilling in dry conditions on aluminum Al 7075-T6. Detection is classified, in this case, as admissible and non-admissible burr, considering industrial tolerances for this process.

### 2.3. Machine Learning Techniques and Unbalanced Industrial Data

As mentioned in Section 1, the datasets extracted from real workshops use to be strongly unbalanced. There are very few works that take into account this fact. Bustillo and Rodríguez [16] showed that pre-processing techniques should be applied to industrial datasets, such as Synthetic Minority Over-Sampling Technique (SMOTE) [27] or undersampling, before a standard machine-learning algorithm can properly model the breakage detection of multitooth tools in real crankshaft machining lines. Martin-Diaz et al. [28] identified a similar limitation of machine-learning algorithms when modeling early fault detection in induction motors. They proposed the use of optimized sampling techniques in the industrial dataset before applying AdaBoost ensembles to model this process.

## 3. Data-Acquisition Set Up

The IoT solution developed in this investigation was, at the machine level, composed of a data-acquisition system connected to the machine control, in this case, a five-axis machining center equipped with Heidenhain 640 CNC. The machining center milling head was equipped with a Pt100 temperature sensor. It had two continuous rotary axes specially designed for machining aeronautical components. The other three axes were the transversal longidinal ones X, Y, Z. It can perform multiple operations through a CNC with very little human intervention. These operations use cutting and rotating tools, such as mills and drills. This data-acquisition system transfers the collected data to a database through an accessible Internet connection in the workshop. The database will oversee automatic and periodic analysis of data collected in real-time. Figure 1 schematically shows the operation of the IoT design solution. Note that only read operations are performed, so the process will never alter the operation of the machine.

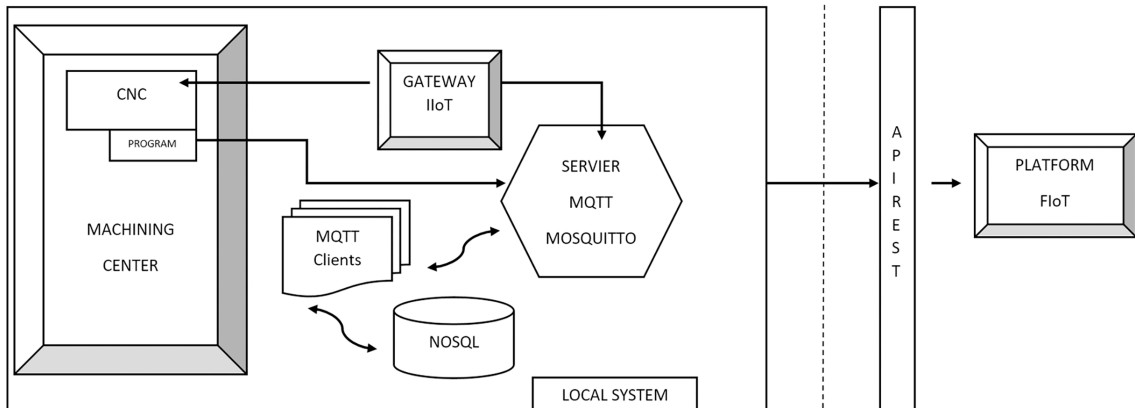

**Figure 1.** Outline of the design of the IoT solution. This schematic diagram shows the operation of the IoT solution. Note that only read operations are performed, so the process will never alter machine operation.

Communication between the acquisition system and the machining center is carried out through an automatic CNC variable reading process. Developed in Python 2.7, information may be captured with this process, using the internal variable name (mnemonic) used in the PLC. The Python library "pyjh" [29] (which is a proprietary software from Heidenhain 640 CNC) was used to read the CNC variables. The equipment can monitor all the parameters of the CNC of the machining center at variable sampling frequencies with which data can be compiled on the general dynamics of the machine, its consumption, its thermal evolution, significant averages of vibrations and shocks and the entire alarm table that occurs.

The output of the reading process of the CNC variables was performed using the "Internet of Things" protocol, MQTT (Message Queue Telemetry Transport), establishing communication between the system and any IoT platform that supports the MQTT protocol. The controller of the machining center must be on the same network as the IIoT Gateway (i.e., Industrial IoT Gateway). According to [30] an IoT Gateway "is a connecting link between the sensor network and the traditional communication network". Nowadays, IIoT Gateways can perform many other tasks, besides their function as a mere protocol converter, such as for example encrypting, buffering, and preprocessing the information. In our case, this device (1) reads the information from the CNC using TCP/IP sockets, and (2) "monitors" it (i.e., builds a JSON (JavaScript Object Notation) object with this information to be published by an MQTT client). There are another two main MQTT clients in this architecture: one to store these JSON objects in a database, and another open to further uses (e.g., plotting graphical information).

The acquired data were recorded in real-time on a private MQTT Mosquitto server in the machining workshop. This server was operated through the publication and subscription system. Each monitoring device has a publication topic within a hierarchical structure that can be used to subscribe to one or more information topics. The MQTT Broker is the server where all requests from MQTT Clients are received. In addition to the three above-mentioned clients that publish information, other clients preprocess and structure the published data.

The preprocessed data is grouped into different categories: consumption, axes, cutters, mandrel, temperature, etc. for future data analysis using data-mining techniques. The MQTT normalization clients republish the information in the MQTT Broker, to which they are subscribed, but already normalized and structured in another topic. Then, the MQTT Client subscribed to those pre-processed data topics will obtain the data and store it persistently in a local non-relational database (NoSQL) (Figure 1). The use of an NoSQL database is required, because of the large data volumes collected per unit of time. It avoids potential scalability problems and improves the speed of querying and writing to the database. These data volumes are temporarily stored in the control server itself.

Twice a day, the information is sent to the Fiware Internet of Things (FIoT) data-processing platform through a communication interface with REST software architecture. The data volumes included in this work that were collected on the FIoT platform exceeded 1,500,000 datums grouped by category, over 3 months of machining center operations, although the specific study described in the following sections only analyzed a very limited part of this information. In this process, another data filtering operation was performed: eliminating possible captures of erroneous measures and adding extra information for later use. This captured information is tagged with machine identifiers, company, location, date, and origin of the data. JSON annotation was also used for the correct management of this information.

## 4. Modeling

### 4.1. Dataset Description

A dataset was extracted from the data-acquisition platform described in Section 3, considering different sorts of information: from axis position to motor temperature. The dataset therefore included: 5 inputs of the machine programmed position (X, Y, Z, B, and C-axis); 5 inputs of the measured cutting tool position (X, Y, Z, B, and C-axis); 5 inputs of the machine programmed speed (X, Y, Z, B and C-axis); and, 4 motor-temperature inputs (X, Y, Z motors, and milling head H). Besides, two machine labels were also extracted from the PLC: the machining mode and the machine state. The machining state can take four different values depending on whether the machine is in Idle state (i.e, ready to work but without any programed order) (1) ; working (2); on stand-by (3); or stopped (4). The machining mode can take five different values depending whether the machine is: following a machining program or automatic mode (1); controlled manually by means of the machine carriage (2); running a program line by line, controlled by an operator in Manual Data Input or MDI mode (3); in Pass Reference Mode to fix new machine references (4); or, running in Single Block Mode (5). While the modes are standard modes predefined by any machine-tool, the machine states have been classified according to standard proposals [12].

While all position, speed, and temperature-related inputs were considered to be continuous attributes, both machine labels were considered as nominal attributes, because ordering their possible values made little or no industrial sense. Although the original dataset also included a timestamp for each measurement, this attribute was not used for training and validation of the prediction model. From this dataset, machine temperature and machine label were considered as the outputs. In the case of temperatures, as they can take any value within a continuous range, the prediction task is called regression; while in the case of machine labels, as these outputs can only take a very limited number of values, the prediction task is called classification (Table 1).

**Table 1.** Dataset attributes and output with their variation range. The inputs and outputs and their abbreviations in the dataset are summarized; the output variables are outlined in bold.

| Variable | Abbreviation |
|---|---|
| Axis Machine programmed position (X, Y, Z, B and C-axis) | $Axis_{X,Y,Z,B,C}$ |
| Cutting tool measured position (X, Y, Z, B and C-axis) | $Tool_{X,Y,Z,B,C}$ |
| Machine speed (X, Y, Z, B and C-axis) | $Speed_{X,Y,Z,B,C}$ |
| **Motor temperature (X, Y, Z motors and milling head H)** | $\mathbf{T_{X,Y,Z,H}}$ |
| **Machining mode** | **Mode** |
| **Machine state** | **State** |

The dataset included 52,592 instances recorded over a period of 3 months. But, as previously outlined in the Introduction, real workshop conditions mean that the datasets are often unbalanced. In this case, the instances distribution between the four different states was as follows (as summarized in Table 2): 34.75% in an idle state; 55.40% in a working state; 0.89% in a stand-by state; and, 8.95% in the stop state. The instance distribution between the four different machine modes was as follows:

74.00% in automatic mode; 13.37% in manual mode; 10.38% in MDI mode; 1.33% in Pass mode; and, 0.93% in Single Block Mode. These proportions seem to be reasonable, considering the daily workload of the workshop: the machine is mainly run in automatic mode in a working state, but the allocation and fixation of the workpieces and their final validation is also in manual mode on the machine (usually measuring the workpiece dimension and allocation with probes) when in the idle state or the stand-by mode. Although the instance distribution might be reasonable from an industrial point of view, it is strongly unbalanced in terms of its data analysis and this fact will affect the accuracy of the machine-learning techniques and the selection of the most suitable metrics to evaluate the performance of the model, as will be discussed in Section 4.2. The temperature distribution, although continuous variables, showed a similar behavior: most of the measured values (around 75%) were close to the programed temperature (25–27 °C), while fewer than 25% of the measurements belonged either to the warming-up stage (23–24 °C) or to the hot temperatures outside the controlled range (28–54 °C).

**Table 2.** Distribution of class values in the two classification problems.

| State | No. of Instances | % | Mode | No. of Instances | % |
|-------|------------------|-------|------|------------------|--------|
| 1 | 18.288 | 34.75% | 1 | 38,938 | 74.00% |
| 2 | 29.124 | 55.40% | 2 | 7,034 | 13.37% |
| 3 | 468 | 0.89% | 3 | 5,463 | 10.38% |
| 4 | 4.712 | 8.95% | 4 | 700 | 1.33% |
| | | | 5 | 487 | 0.93% |

*4.2. Machine-Learning Techniques and Best Metrics*

4.2.1. Classification

The metric usually used for classification problems is the percentage of test instances in which the trained model correctly predicts the class. However, in unbalanced problems this metric will not properly evaluate the models that are obtained, since, for example, a classifier that always predicts mode 1 would always achieve a 74% success rate.

There are several metrics for unbalanced classification, but many of them are targeted at two-class problems. For multi-class problems, it is usual to use the Matthews correlation coefficient (i.e., MCC), and the F-measure in both its micro version (i.e., F-micro) and its macro version (i.e., F-macro).

The Matthews Correlation Coefficient, or MCC, is a state-of-the-art metric for unbalanced problems that has a multi-class variant [31]. To compute it, the confusion matrix must be previously calculated. This is a square matrix in which the number of rows and columns correspond to the number of classes in the problem. Therefore, the elements in the diagonal of the matrix represent the cases in which the classifier predicts the correct class. Once the matrix is obtained, the MCC can be calculated as:

$$MCC = \frac{\sum_k \sum_l \sum_m C_{k,k} C_{l,m} - C_{k,l} C_{m,k}}{\sqrt{\sum_k (\sum_l C_{k,l})(\sum_{k'|k' \neq k} \sum_{l'} C_{k',l'})} \sqrt{(\sum_k (\sum_l C_{l,k}))(\sum_{k'|k' \neq k} \sum_{l'} C_{l',k'})}} \tag{1}$$

The F-Measure metric, for the binary case was calculated from the Precision (Pr) and Recall (Re) metrics.

$$P_r := \frac{TP}{TP + FP} \tag{2}$$

$$R_e := \frac{TP}{TP + FN} \tag{3}$$

where:

- TP (i.e., true positives): number of times the classifier correctly predicts the minority class.

- FP (i.e., false positives): number of times the classifier incorrectly predicts instances of the majority class as instances of the minority class.
- FN (i.e., false negatives): number of times the classifier is wrong when predicting minority class instances as majority class instances.

Precision is therefore the ratio of the number of instances of the minority class to the instances predicted as belonging to the minority class; while recall is the ratio of the number of instances predicted as belonging to the minority class to the size of that class.

The F-measure metric combines both measures according to the following equation:

$$F_{measure} = 2 * \frac{P_r * R_e}{P_r + R_e} \tag{4}$$

Hence, the F-Measure is the harmonic mean of both magnitudes. By expanding Equation (4), another expression can be found for the F-measure.

$$F_{measure} = \frac{2 * TP}{(2 * TP + FP + FN)} \tag{5}$$

There are two variants to fit this metric to the multi-class case: F-Macro average and F-Micro average.

The F-Macro computation sets up a binary classification problem for each class, where each class acts as a minority class versus an artificial class consisting of the union of all the other classes (i.e., OVA = One vs. All). Then, for each of these problems the F-measure is calculated, and by averaging all these F-measures the F-Macro is finally obtained.

In the case of the F-Micro, we obtain for each class the values of TP, FP, and FN. All the TPs are added up to obtain a global TP. A global FP and FN could be obtained in the same way. Then, Equation (5) is applied with the three global values, and the F-measure then obtained is the F-Micro.

Forman et al. [32] reported some situations where a division by zero can be achieved in the F-Macro calculation using cross validation. Namely:

1. When using Equation (2) and $TP + FP = 0$, resulting in no definition of precision.
2. When also using equation Equation (3) and $TP + FN = 0$, resulting in no definition of recall.
3. When using equation Equation (5), $TP + FN = 0$ (i.e., no positive instances on the test partition) and also $TP + FP = 0$ (i.e., the classifier predicts no instance as positive).

Because the experiments were conducted using WEKA [33], cases 2 and 3 are impossible as, by default, it uses a stratified cross-validation. Not so with case 1, where WEKA returns a *NaN*, causing these partitions to be ignored when analyzing the results.

It is recommended that in all these cases the resulting F-macro be considered zero [32], otherwise the F-macro obtained by ignoring these situations would be too optimistic. In the experiments, WEKA was reprogrammed to adopt the latter behavior, since situations of this type were present in the datasets under evaluation.

The follow methods were used in the experiment over classification problems :

1. Naïve Bayes [34] which classifies by assigning probabilities to each class using Bayes theorem. It is taken as a baseline to compare with the other classifier methods, due to its simplicity, so methods with worse results than Naïve Bayes would not be acceptable.
2. kNN [35] calculates the distance to all instances from the instance to predict (Euclidean distance was used in the experiments). The majority class of the closest k instances is predicted. The value of k is a parameter of this algorithm. For each cross-validation partition in the experiments, the optimal k value from the integers 1 to 10 was used.

3. Decision trees. J48 was used, which is the WEKA implementation of the C4.5 decision tree [36]. The branching criterion of C4.5 is the information gain.

4. Function-based methods, such as Support Vector Machines (SVM) and Neural Networks:

(a) The Multilayer Perceptron [37] is a neural network that has a number of hidden layers of neurons. The connections between the neurons have a weight which is obtained from a backpropagation algorithm. In the experiments, only one hidden layer was used. The number of neurons in this layer was given by the heuristics (number of independent variables +1)/2.

In the Multilayer Perceptron, the learning rate and momentum parameters were optimized through internal cross validation, in order to maximize the F-Macro. The WEKA Multisearch package was modified for this purpose, as it is not prepared for optimization by F-Macro.

(b) Support vector machines or SVMs [38] are actually classifiers for binary problems. They calculate a hyperplane that separates the regions of space corresponding to two classes. This hyperplane is said to maximize the margin, which intuitively means that it is as far as possible from the points on the border of both regions. These points are the so-called support vectors.

Therefore, for the SVM to work properly, the regions corresponding to both classes need to be linearly separable, which is not too often the case. There are two strategies to adapt the algorithm to problems that are not linearly separable :

i. A parameter C is introduced in the algorithm. It represents how much the incorrect classification of training instances (i.e., the ones falling on the wrong side of the hyperplane) are penalized by the margin optimization procedure.

ii. A transformation of the classification problem from the original features space to another space by means of a non-linear transformation. In that new space, it is expected that the problem will be linearly separable, or at least, the number of instances that can cause that lack of linear separability will be reduced. To reach this goal, the concept of kernel is introduced. A kernel is a special type of function that computes the scalar products between instances in the new space. This scalar product is necessary for the calculation of the hyperplane. The most popular kernel is the Radial Basis Function kernel (SVM-RBF), which is given by Equation (6).

$$K(x_i, x_j) = e^{-\frac{\|x - x_i\|^2}{\gamma^2}} \tag{6}$$

The letter $\gamma$ represents the bandwidth, and it is a parameter of the method.

When an SVM does not use a kernel, it is said to be an SVM with a linear kernel. Whether or not a linear kernel is used, the C parameter is present and must be specified.

Adapting SVM to multi-class problems is done using the 1 vs. 1 strategy, whereby a problem of n classes becomes (n−1)! binary problems. These problems result from confronting each class with all the rest. For each binary problem, an SVM is then trained, and the final prediction comes from the majority vote of all those SVMs.

Two implementations of SVM were used in the experiments:

- LibSVM [39]: used to build Radial Basis Function kernel SVMs.
- LibLinear [40]: used to compute linear kernel SVMs.

For LibLinear, the C parameter was optimized, and in the case of LibSVM, the $\gamma$ parameter was also optimized. In all cases, cross validation was used on the training set. The aim of both optimizations was to maximize the F-macro, using the same modification of the WEKA Multisearch package described above for the Multilayer Perceptron.

5. Bagging-based methods [41] in which several decision trees are created. These trees are different because they are trained by resampling the training set. The final prediction in these ensembles is the one obtained by the majority vote of each of these trees. The trees used for the ensemble are commonly referred to as base classifiers.

   The Bagging variants used in the experiment were:

   (a) Bagging using C4.5 trees as base classifiers

   (b) Bagging using Random Balance of C4.5 trees [42]. Random Balance changes the number of instances of each class before they are used by each C4.5. Rather than balancing the number of instances of each class, this technique assigns a random number of instances to each. To do so, it both resamples the training set when it needs less instances, and it also creates synthetic instances as and when it needs. To create these synthetic instances, it uses SMOTE [27]. Random Balance is oriented towards unbalanced multiclass datasets, so it is *a priori* an approach that fits the nature of the problem under analysis very well.

   (c) Random Forest [43] can be considered a Bagging technique in which the base classifiers are "Random Trees". In these decision trees, each time a branching node is to be built, the possible attributes to be considered are randomly restricted. In WEKA, Random Trees are implemented as a modification of REP-Trees (i.e., Reduced Error Pruning trees), which have a fast building process.

   (d) Rotation Forest [44]. In this method, before building each tree, and once the training set has been resampled, the features are grouped randomly. These groups are disjoint from each other; and the union of all the groups contains all the attributes. All groups have the same number of attributes n, except perhaps one of the groups, when the dataset number of attributes is not divisible by n. In each group, the PCA projection is computed, taking only a number of principal components (i.e., n in most cases) to keep the size of the groups unchanged.

   In the experiments, Rotation Forest of C 4.5 was used, as was Rotation Forest of Random Forest, because its training time is significantly shorter with the same number of trees.

   All Bagging configurations have been computed using 100 trees. For this reason, Rotation Forest of Random Forest calculates 10 Random Forests, each containing 10 trees.

6. Finally, two Boosting based ensembles were included for classification:

   (a) AdaBoost.M1 [45], because it is the most popular Boosting ensemble for classification. In this ensemble each base classifier is derived from the previous one, so that it gives more weight to the instances that the base classifier of the immediately preceding iteration has incorrectly classified. Unlike Bagging, the final prediction is not made by a majority vote of the base classifiers, but by a vote weighted by the individual error of each one. In the experiments, 100 C4.5 trees were taken as the base classifiers.

   (b) LogitBoost [46], because it is more suitable than AdaBoost.M1 for working with multi-class problems. This method makes a logistic transformation to convert the classification problem into a regression problem that predicts the probability of the instance belonging to each class. Therefore, LogitBoost uses regressors (i.e., continuous value predictors), instead of classifiers, as base predictors. LogitBoost uses an additive regression in each iteration by appending regressors that learn the residues of the probability predictions (i.e., they learn the differences between the predicted values and the values of the training

set). In each iteration, the probabilities of each training instance can be estimated by adding the predictions of the residuals. With that estimation, weights may be given to the instances, to add more importance to those where a higher prediction error occurs. The base regressors used in the experiments were REPTree regression trees. The size of the ensemble is 100 iterations as in the other cases.

### 4.2.2. Regression

The metric used for regression problems is the Root Relative Squared Error (RRSE). Let $y_i$ be each of the values to be predicted in the test set, let $p_i$ be the predictions that a regression method outputs for those values, and let $\overline{y}$ be the mean value of $y_i$ in the test set. RRSE is then defined as follows:

$$\sqrt{\frac{\sum(y_i - p_i)^2)}{\sum(y_i - \overline{y})^2}} \tag{7}$$

Therefore, the values of this metric are always greater than, or equal to, zero. In the ideal case of a regressor predicting without error, the RRSE would be zero. In a naive regressor that would always predict the mean $\overline{y}$, both the numerator and the denominator of Equation (7) would be equal, then the resulting RRSE value will be 1 (or 100% if expressed as a percentage). The reason for having chosen RRSE as a metric is precisely because it provides a figure that takes this naive regressor as a baseline. A regressor could have an RRSE greater than one, although it would obviously be of no practical interest.

The regression methods used in the experiment were:

1. Function-based methods such as Linear Regression, Support Vector Machines (SVM), and Neural Networks:

    (a) Linear Regression creates a function that is a linear combination of the input variables, which minimizes the quadratic error of the training set. The Akaike criterion, as the WEKA implementation default, is used to select the attributes of the model.

    (b) The simplest version of SVM for regression is the linear SVM [47]. As in the case of linear regression, we also have a model that is a linear combination of the input variables (i.e., a hyperplane), but this time the optimization process ignores the instances that are less than $\epsilon$ away from that hyperplane (i.e., they are within the "margin"). The distance, $\epsilon$, is a parameter to specify that is denoted by C in most implementations. The implementation used in the experiment was LibLinear [40], for which the C parameter was optimized by cross validation, to minimize RRSE, using WEKA's Multisearch package.

    The SVM for regression, like its version for classification, can use a kernel, and generate the hyperplane in a different space, leading to a more complex geometry than in the original space, which is more appropriate if the behavior of the variable to be predicted is not linear. Therefore, in the experiment we have also used an SVM regressor with a Radial Basis Function kernel (SVM-RBF). In the case of this kernel, besides optimizing C, it is necessary to optimize the bandwidth parameter. As in other methods, Multisearch was also used, using internal cross validation and aiming at minimizing the RRSE. The SVM-RBF implementation used in the experiment is the regression version of LibSVM [39].

    (c) The neural network method used is the same as that described for classification (i.e., a Multilayer Perceptron [37]). As in the classification, only an intermediate layer was used. The number of neurons in this layer is also given by the heuristics (number of independent variables + 1)/2. Likewise, the learning rate and momentum parameters were optimized with Multisearch, again through internal cross validation, to minimize the RRSE.

2.  kNN [48] works similarly to kNN for classification. The difference is that, once the k instances closest to the test instance are found, the class mean for those k instances is predicted. As for classification, the value of k was also optimized in the experiments for each cross-validation partition with the best value obtained for k ranging between 1 and 10.

3.  Regression trees. Two types of decision trees for regression were used:

    (a)  M5P [49] which is a type of tree in which a linear regression is created with the instances in each leaf, so the prediction that is returned is the one of the linear regression corresponding to the leaf in which the test instance falls.

    (b)  REP-Tree [33] is WEKA's "native" decision tree for both classification and regression. When used as a regressor, it returns the mean value of the feature to be predicted for the instances of the leaf with the test instance. REP-Tree uses variance reduction as a criterion for finding split points for each node. The tree is pruned using the Reduced Error Pruning technique.

    M5P is not in the results tables of the next section, because values higher than 1 were always obtained for RRSE, both when it was used alone and when it was tested within some variants of ensembles for regression.

4.  Methods based on ensembles [50], which combine the prediction of other simpler regressors called base regressors. The base regressors used in the experiments were REPTrees, since M5Ps always led to RRSE greater than 1. The ensembles used in the experiment were:

    (a)  Bagging [41]. In this ensemble each base classifier is trained with a sample of the training set with replacement. The sample is the same size as the training set. The prediction is the average of the predictions of the base regressors.

    (b)  Iterated Bagging [51]. In this method the first iteration applies Bagging over the original training set. The differences (i.e., residuals) between the training values predicted by that initial Bagging and the actual values are then computed. In the next iteration, Bagging is done again, but this time predicting the residuals. At this point the ensemble prediction would be obtained from the sum of the predictions of both Baggings. With this prediction, a new set of residuals are calculated, which are used to train another new Bagging, and so on during n iterations. In the experiments 10 iterations of Baggings of 10 regression trees were used.

    (c)  Rotation Forest for regression [52] is a variant of Bagging in which the base regressors are transformed in the same way as in the Rotation Forest variant for classification. That is, random and discrete groups of features are created and the PCA projection is applied to each group.

    All the ensembles have 100 base regressors. That is the reason why Iterated Bagging was configured with 10 iterations of Baggings of 10 trees.

## 5. Results and Industrial Interpretation

A dataset has been created for each of the six variables to be predicted. All these sets have the same input variables, and have a single output variable, which is different in each of them.

For both classification and regression problems, the methods described in the previous section were applied to each of the datasets using $10 \times 10$ stratified cross-validation. For each of the metrics, the average values were found in the 10 repetitions $\times$ 10 partitions. The corrected re-sampled *t*-test was used [53] at a confidence level of 95%, in order to determine whether there were significant differences in the results for each metric between two methods applied to the same data set.

*5.1. Classification: State and Mode Prediction*

Table 3 shows the 13 classification methods tested in the experiment. Each method is compared with the other 12 methods for the 2 classification problems State and Mode (i.e., $12 \times 2 = 24$ comparisons). One method for each one of these 24 comparisons can receive a significantly better result than the other method. So, in this case, we say that the method "wins". The method can also get a significantly worse result than the other method, and then we say that it "loses". Finally, the comparison may return no significant differences.

In Table 3, columns **V** and **D** respectively represent the number of these significant wins and losses in the two classification problems for the method in that row versus all other methods. Then, the difference "Δ" is computed by subtracting D from V. This difference is taken as an indicator of what the best method is. In Table 3, the methods are ordered by Δ. We can see from the table that the two best positioned methods are tied with Δ equal to 16. They win 16 times from the 24 matches, and they never lose.

The two rightmost columns of Table 3 contain the average F-Macro values for each method and predicted variable. These average values are computed from the 100 experiments from the $10 \times 10$CV. When these values are followed by "*", they are significantly worse than the one achieved by the first Δ ranked method (i.e., Rotation Forest of Random Forest in the table, but we also tested that "*"s keep unchanged, if the best method was considered Random Forest). The best F-Macros for each predicted variable are highlighted in bold (i.e., LogitBoost for State and Bagging of Random Balance of C4.5 for Mode). We also tested that "*"s in the State column keep unchanged, if the methods were compared to LogitBoost, as well as in the Mode column, if the methods were compared to Bagging of Random Balance.

**Table 3.** Ranking of F-Macro in the classification problems. V is the number of times the method has a better statistically significant F-Macro when compared with the other 12 methods through the 2 predicted variables. D is the number of times the method has a worse statistically significant F-Macro when compared with the other 12 methods through the 2 predicted variables. The ordering criteria Δ, is equal to V – D. The columns State and Mode have the average F-Macro figures for the method using $10 \times 10$CV. (*) represents this method has an average F-Macro that is significantly worse than the one for the best-ranked method. The best F-Macros for State and Mode are highlighted in bold.

| Δ | V | D | Method | State | Mode |
|---|---|---|--------|-------|------|
| 16 | 16 | 0 | Rotation Forest of Random Forest | 0.99342 | 0.92407 |
| 16 | 16 | 0 | Random Forest | 0.99283 | 0.92444 |
| 15 | 16 | 1 | LogitBoost of REPTree | **0.99370** | 0.92056 |
| 14 | 15 | 1 | AdaBoost.M1 of C4.5 | 0.99297 | 0.92297 |
| 10 | 15 | 5 | Bagging of Random Balance of C4.5 | 0.98857 * | **0.92785** |
| 8 | 13 | 5 | Rotation Forest of C4.5 | 0.99298 | 0.91275 * |
| 2 | 11 | 9 | Bagging of C4.5 | 0.99115 * | 0.91359 * |
| −3 | 9 | 12 | C 4.5 | 0.99056 * | 0.90944 * |
| −6 | 8 | 14 | kNN | 0.98119 * | 0.90899 * |
| −13 | 5 | 18 | Radial Basis Function SVM | 0.82836 * | 0.77250 * |
| −15 | 4 | 19 | Multilayer Perceptron | 0.79272 * | 0.66775 * |
| −21 | 1 | 22 | Naïve Bayes | 0.64369 * | 0.44891 * |
| −23 | 0 | 23 | Linear SVM | 0.64819 * | 0.39074 * |

The following conclusions may be identified from Table 3:

1. Very good results may be found with just one single C4.5 decision tree. It could, on the one hand, mean that the input variables adequately describe the output variables and, on the other hand, that the number of instances is also sufficient.
2. Both problems are not suitable for a linear classification in view of the results of the linear SVM.

3.  Classifiers based on the optimization of a complex function, such as Neural Networks or Radial Basis Function SVM, do not obtain competitive results despite the computational cost involved in the parameter optimization process.
4.  All the top ranked methods are ensembles, which hardly vary from each other in their performance.
5.  kNN is the best of the non-ensemble methods. It is once again due to the low number of characteristics (i.e., there is no curse of dimensionality) and the high number of instances.

Conclusions 2 and 3 point out the complexity of the decision boundaries for both problems.

Tables 4 and 5 repeat the same structure as Table 3, but using F-Micro and MCC. In Table 4, the best F-Micro for State is achieved by LogitBoost (i.e., not by the best ranked method), although we checked that "*"s remained the same in the State column, if the methods were compared to LogitBoost. However, in Table 5, the best MCC value for the Mode variable is achieved by the second ranked method (Bagging of Random Balance of C4.5). In this case, when the methods were compared to this Bagging variant, every other method would be tagged with "*". So, when MCC was chosen as a metric, Bagging of Random Balance of C4.5 was significantly better than all the other methods for the Mode variable.

**Table 4.** Ranking of F-Micro in the classification problems. V is the number of times the method has a better statistically significant F-Micro when compared with the other 12 methods through the 2 predicted variables. D is the number of times the method has a worse statistically significant F-Micro when compared with the other 12 methods through the 2 predicted variables. The ordering criteria $\Delta$, is equal to V − D. The columns State and Mode have the average F-Micro figures for the method using $10 \times 10$CV. (*) represents this method has an average F-Micro that is significantly worse than the one for the best-ranked method. The best F-Micros for State and Mode are highlighted in bold.

| Δ | V | D | Method | State | Mode |
|---|---|---|---|---|---|
| 20 | 20 | 0 | Rotation Forest of C4.5 | 0.99756 | **0.94374** |
| 13 | 16 | 3 | Bagging of C4.5 | 0.99711 * | 0.94292 |
| 12 | 15 | 3 | LogitBoost of REPTree | **0.99759** | 0.94125 * |
| 11 | 14 | 3 | AdaBoost.M1 of C4.5 | 0.99755 | 0.93946 * |
| 10 | 15 | 5 | C 4.5 | 0.99696 * | 0.94307 * |
| 10 | 13 | 3 | Rotation Forest of Random Forest | 0.99752 | 0.94007 * |
| 3 | 10 | 7 | Random Forest | 0.99710 * | 0.94029 * |
| 1 | 10 | 9 | Bagging de Random Balance of C4.5 | 0.99664 * | 0.93855 * |
| −8 | 8 | 16 | kNN | 0.99198 * | 0.93582 * |
| −12 | 6 | 18 | Multilayer Perceptron | 0.93112 * | 0.90334 * |
| −16 | 4 | 20 | Radial Basis Function SVM | 0.88398 * | 0.88322 * |
| −22 | 1 | 23 | Linear SVM | 0.80947 * | 0.71559 * |
| −22 | 1 | 23 | Naïve Bayes | 0.81845 * | 0.51293 * |

Some additional conclusions that could be drawn from these two tables are:

1.  The C4.5 tree reached very similar values to those of some ensembles. Usually, it is expected that the variance component of the classification error decreases as the training dataset increases [54]. It is known that Bagging-based ensembles reduced this error component, as Boosting-based ensembles also do in their later iterations [55]. Hence, these similar results of C4.5 vs. ensembles can be explained, at least in part, by the dataset size and the superiority of decision trees over the other non-ensemble methods.
2.  It is also interesting to note that the F-Micro and MCC for State variable prediction almost reaches one (i.e., the maximum possible value) in many of the classifiers that were tested, and in nearly all in the case of F-Micro. This points out again that the State is well characterized by the attributes of this data set.
3.  However, the Mode variable figures are worse. For both F-Macro and MCC metrics, Random Balance Bagging, which is a specific method for imbalanced datasets, is the best choice. The "+" sign in

this method in Table 5 indicates that the MCC obtained is significantly better than the first method in the ranking.

These last 2 conclusions can be explained by the fact that the Mode variable has a much clearer majority class (i.e., *Mode* = 1 has 74% of the instances); while the State variable is distributed between states 1 and 2 in a quite balanced way for 90% of the instances.

**Table 5.** Ranking of MCC in the classification problems. V is the number of times the method has a better statistically significant MCC when compared with the other 12 methods through the 2 predicted variables. D is the number of times the method has a worse statistically significant MCC when compared with the other 12 methods through the 2 predicted variables. The ordering criteria Δ, is equal to V – D. The columns State and Mode have the average MCC figures for the method using $10 \times 10$CV. (*) represents this method has an average MCC that is significantly worse than the one for the best-ranked method. (+) represents this method has an average MCC that is significantly better than the one for the best-ranked method. The best MCC for State and Mode are highlighted in bold.

| Δ | V | D | Method | State | Mode |
|---|---|---|---|---|---|
| 17 | 18 | 1 | Rotation Forest of C4.5 | **0.99596** | 0.85325 |
| 12 | 17 | 5 | Bagging of Random Balance of C4.5 | 0.99406 * | **0.86118**+ |
| 12 | 14 | 2 | Bagging of C4.5 | 0.99504 * | 0.85069 |
| 10 | 14 | 4 | C 4.5 | 0.99475 * | 0.85193 |
| 10 | 13 | 3 | LogitBoost of REPTree | 0.99591 | 0.84690 * |
| 9 | 13 | 4 | AdaBoost.M1 of C4.5 | 0.99579 | 0.84381 * |
| 8 | 12 | 4 | Rotation Forest of Random Forest | 0.99569 | 0.84476 * |
| 2 | 10 | 8 | Random Forest | 0.99483 * | 0.84530 * |
| −8 | 8 | 16 | kNN | 0.98624 * | 0.83288 * |
| −12 | 6 | 18 | Multilayer Perceptron | 0.87607 * | 0.76677 * |
| −16 | 4 | 20 | Radial Basis Function SVM | 0.81216 * | 0.68454 * |
| −21 | 2 | 22 | Naïve Bayes | 0.64746 * | 0.29481 * |
| −23 | 0 | 23 | Linear SVM | 0.62066 * | 0.26261 * |

The computation of confusion matrices for some of the methods can provide deeper insight into their accuracy. In a confusion matrix, the element in row *i* and column *j* represents the number of times the class in row *i* is predicted by the method as the class in column *j*.

In an ideal classifier that is correct 100% of the time, the elements of the diagonal will achieve 100% values (i.e., the elements of class *i* will always be correctly predicted as belonging to class *i*). Also, the rest of the cells within that ideal classifier that lie outside the diagonal will have a value of 0% (i.e., the elements of class *i* would never be predicted as belonging to a different class *j*).

A $5 \times 2$ CV was performed (i.e., 5 repetitions of 2 folds cross validation), to produce reliable confusion-matrix values. Hence, the data set for a $5 \times 2$ CV was randomly divided into 2 parts at 50%. One part was used to train the classifier and the other, to validate the model by counting which class it rightly or wrongly predicted for each test instance. Then, the partitions were swapped (i.e., The partition that was previously used for training was then used for validation and vice-versa). The 50% random split differed for each of the 5 repetitions. Once that procedure had finished, $5 \times 2 = 10$ values for each cell of the matrix were obtained. The figures on the following page represent the average values of those 10 results. The data are shown as percentages instead of absolute values, so that minority classes are not underrepresented.

Figure 2 shows the confusion matrices for the Mode classification problem. The two upper matrices represent the best methods in that problem (i.e., Bagging of Random Balance, because it has the highest values for F-Macro and MCC, and Rotation Forest of C4.5, because it has the highest value for F-Micro). The two lower matrices represent the two best non-ensemble methods (i.e., a C4.5 decision tree and k-NN). All the classifiers show diagonal values above 90%, except for *Prog Line* class, which is confused half the time with *Automatic* class. This confusion can be explained, as the first time that a new machining program is executed, the machine operator will usually run it line by

line, so that its performance can be closely controlled. In this way, the operator can quickly stop the program execution, if a program error is detected that might damage the workpiece or the cutting tool. Once the program has been validated by machining a first workpiece, automatic mode will be used to run this program in the future. Therefore, the machine-learning algorithm will find no difference between automatic mode and line by line mode in those cases. However, Bagging of Random Balance performs well even for that fuzzy case.

In contrast, Figure 3 showed high performance in all scenarios with any of the classifiers of the figures.

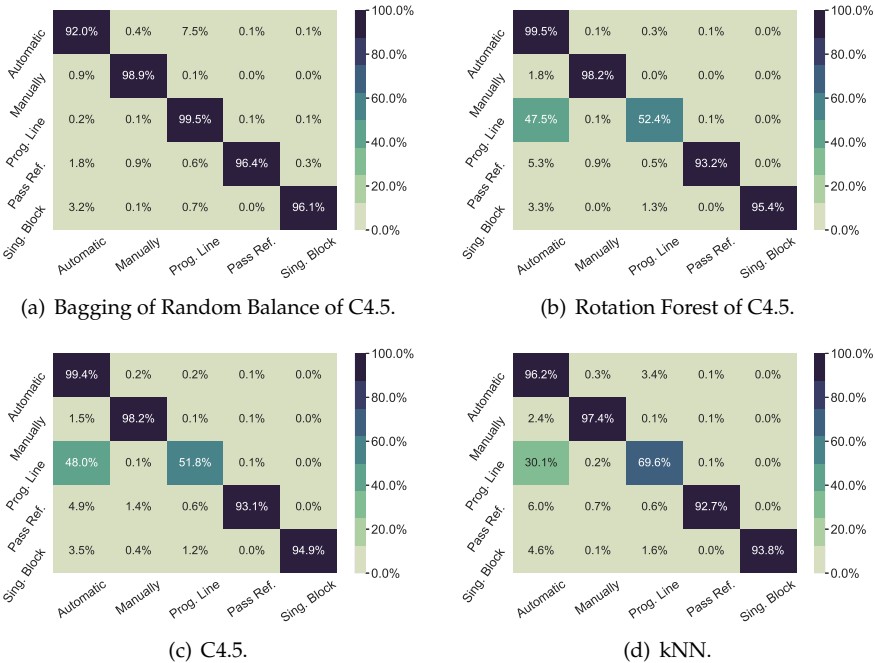

**Figure 2.** Confusion matrix over *Mode*.

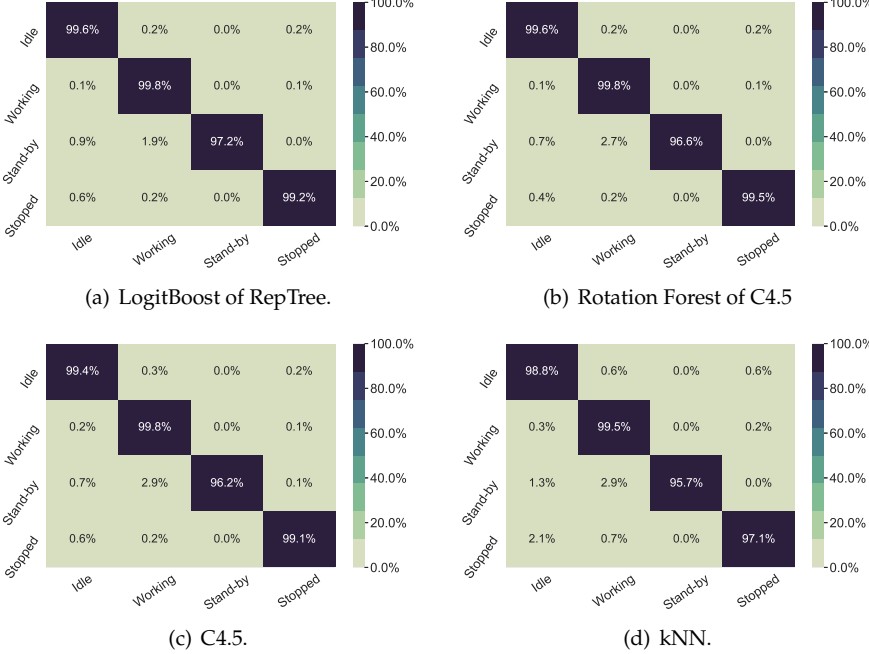

**Figure 3.** Confusion matrix over *State*.

In short, the high performance that Bagging of Random Balance achieved for Mode prediction, and LogitBoost for State, will guarantee that this model can identify these two process parameters. The combination of both can surely identify the main production activity of the machining center. Some examples of these combinations are: (1) reference fixation of a new workpiece (idle state and manual mode); (2) workpiece milling operation (working state and automatic mode), test of a new milling program (idle state and MDI mode), and load/unload of a workpiece (stopped state and Pass mode). Therefore, the prediction model can extract useful information on how the machining time is distributed, not from an Engineering Department planning report, but from the real workshop reports on the daily situation.

## 5.2. Regression

As for regression experiments (Table 6), there is one method that significantly beats all others in all the variables to be predicted, which is Rotation Forest of REP-Trees.

**Table 6.** Ranking of RRSE in the regression problems. V is the number of times the method has a better statistically significant RRSE when compared with the other 9 methods through the 4 predicted variables. D is the number of times the method has a worse statistically significant RRSE when compared with the other 9 methods through the 4 predicted variables. The ordering criteria $\Delta$, is equal to $V - D$. The columns $T_H$, $T_x$, $T_y$ and $T_z$ have the average RRSE figures for the method using $10 \times 10CV$. (*) represents this method has an average RRSE that is significantly worse than the one for the best ranked method. The best RRSEs for each predicted variable are highlighted in bold.

| $\Delta$ | V | D | Method | $T_H$ | $T_x$ | $T_y$ | $T_z$ |
|---|---|---|---|---|---|---|---|
| 36 | 36 | 0 | Rotation Forest of REP-Tree | **55.072** | **61.526** | **60.957** | **64.021** |
| 28 | 32 | 4 | Bagging of REP-Tree | 55.582 * | 62.668 * | 62.208 * | 65.121 * |
| 20 | 28 | 8 | Iterated Bagging of REP-Tree | 56.124 * | 63.446 * | 62.892 * | 65.952 * |
| 11 | 23 | 12 | REP-Tree | 58.935 * | 67.659 * | 67.012 * | 70.949 * |
| 5 | 20 | 15 | k-NN | 64.560 * | 68.794 * | 67.817 * | 71.014 * |
| −7 | 13 | 20 | Radial Basis Function SVM | 78.018 * | 85.287 * | 85.299 * | 88.811 * |
| −9 | 12 | 21 | Multilayer Perceptron | 79.858 * | 87.524 * | 87.074 * | 90.586 * |
| −22 | 6 | 28 | Linear Regression | 93.251 * | 95.243 * | 94.902 * | 95.927 * |
| −26 | 4 | 30 | Linear SVM | 94.522 * | 95.681 * | 95.800 * | 97.122 * |
| −36 | 0 | 36 | ZeroR (i.e., always predict the average) | 100.000 * | 100.000 * | 100.000 * | 100.000 * |

Again, the ensembles are in the lead. Linear regressors (i.e., Linear Regression and Linear SVM) are left in the tail positions, which suggests that these are not linear problems. As with classification problems, a single decision tree is still the best performing non-ensemble method.

WEKA gives the name ZeroR to the naive regressor that always predicts the average. For that reason, it is at the end, since by definition RRS will always take a value of 100%. ZeroR is taken as the baseline.

The average of the absolute error was calculated for the best method (i.e., Rotation Forest of REP-Tree) and for ZeroR (Table 7), to compute the physical magnitude of the error committed. The absolute error is defined as the absolute value of the difference between the actual value and the predicted value. The table shows average absolute errors ranging from 0.68 °C for $T_H$, to 1.99 °C to $T_z$.

**Table 7.** Average of the absolute error for the best method, and the error that would be committed if the average of $T_H$, $T_x$, $T_y$, $T_z$ were always predicted (i.e., if ZeroR is used).

| Variable | Rotation Forest of REP-Tree | ZeroR |
|---|---|---|
| $T_H$ | 0.68 | 1.43 |
| $T_x$ | 1.40 | 2.49 |
| $T_Y$ | 1.47 | 2.63 |
| $T_Z$ | 1.99 | 3.31 |

For a deeper understanding of temperature behavior, Figure 4 was plotted. Figure 4 shows the box-plot diagram for the four motor temperatures. In the box-plot diagrams the data are split into four quartiles depending on their value: the lower, second, third and upper quartiles, each of them with 25% of the data. The box-plot diagrams show: (1) the median of the variable with a green line; (2) the area with 50% of the middle data (data in the second and third quartiles) with a blue box; (3) two black lines or whiskers; and, (4) the data outside the whiskers with black circles. The whiskers are calculated as 1.5 times the interquartile range (distance between the upper and the lower quartile). Whiskers are a fundamental measure because any data outside the whisker should be considered as abnormal data or outliers [56]. Figure 4 shows that the temperatures do not deviate by more than 2.5 °C from the temperature of the median for any motor. The temperature dispersion, evaluated from the whiskers distance, is approximately the same for the linear axes: those that are most stressed during a machining operation). In the case of the milling head, the temperature dispersion is smaller, as expected in a thermalized milling head. However, Figure 4 also shows:

1.  Many outliers at low temperatures for $T_X$, $T_Y$ and $T_Z$, which may be due to the latency in the system's heating curves at start-up.
2.  Some outliers for the Z-axis at high temperatures. These points may indicate that this axis has been over-worked and strained at certain points during machining, and their identification may be important to avoid damage to the spindle motor or to increase the average life of the tool.
3.  Some outliers for the milling head temperature $T_H$ at high temperatures, but not far away from the median temperature. These values may indicate some rotating efforts of the milling head during 5-axis machining, but not too high to overcome the limit of 35 °C.

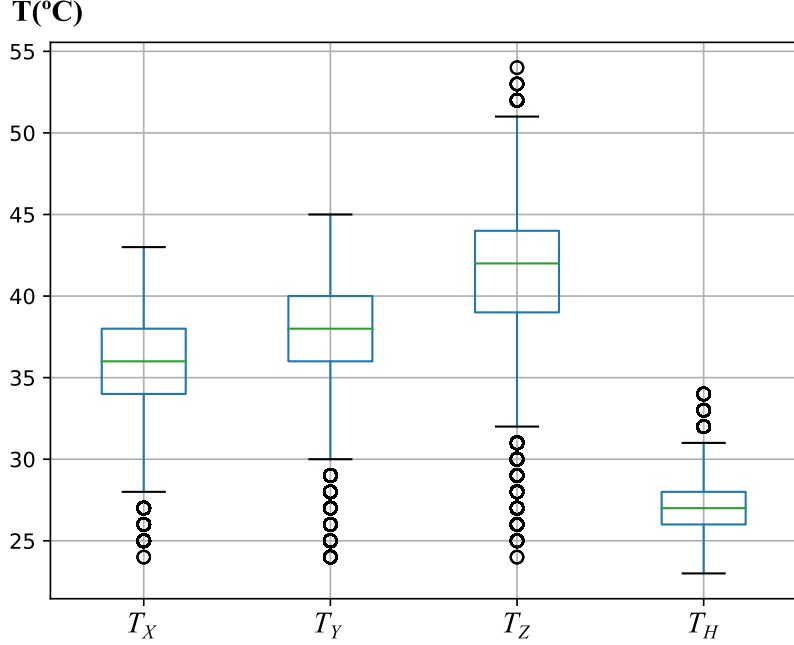

**Figure 4.** Box plots for temperature variables.

The comparison of the errors for ensemble models are shown in Table 7 and the 2.5 °C deviations around the median, in Figure 4, point out that these predictors are useful to detect the outliers, which are of greater interest, because they can refer to overly demanding cutting conditions or abnormal programming of the machine. Moreover, those absolute error values may seem a little high for data close to the median. However, these data instances are the ones with the lowest industrial value, as they are corrected by the continuous action of the cooling system.

## 6. Conclusions

The main contribution of this work may be summarized as follows:

1. A real data-extraction architecture connected to an IoT platform for small workshops has been described.
2. The data that are extracted can be useful for solving industrial problems. High performance results can be achieved for industrial problems related to both imbalanced classification and regression.
3. The best performance was obtained by machine-learning ensemble methods, which require no method optimizations, yielding a straight-forward and simple way for optimal exploitation of the data that were gathered for this study.

The IoT platform with which we have experimented was able to extract many useful inputs from the CNC of the machine without interfering with the production process. In the present work it collected a dataset consisting of measurements of real machining operation during lengthy time periods (3 months, which means 52,592 instances) and the different conditions included in it: milling and drilling processes of many different workpieces, many warm-up cycles, tools with strong wear effects and new tools, etc. Within this period, the machine state and operation mode, the position and speed of its axis and of the cutting tool and the temperature of 4 different components were all measured.

As for the use of the dataset that was obtained with machine-learning models, the present work represents an attempt to see whether these machine-learning techniques will also work properly with real data from machining workshops, to extract useful information on production planning and performance. Two types of industrial problems have been addressed by applying several classification and regression algorithms.

- The prediction of two discrete variables: State and Mode (i.e., two classification problems).
- The prediction of four continuous variables: $T_X$, $T_Y$, $T_Z$, $T_H$ (i.e., four regression problems).

IoT in small workshops will usually have to negotiate with unbalanced and not very reliable data. However, prediction tasks are performed with high accuracy, despite the strongly unbalanced nature of the acquired data. State prediction performed well even with a single decision tree, although, in general, ensembles improved these results. In fact, there are several methods that are close to reach the maximum value of "1" for F-Macro, F-Micro and MCC metrics. On the other hand, Mode prediction is more problematic, and the best results have been obtained with an unbalanced problem-oriented ensemble, such as Random Balance's Bagging.

In the four regression (i.e., temperature prediction) problems, however, there is a clearly better method, as Rotation Forest always outperforms all other methods significantly. This algorithm is of limited accuracy when predicting the favorable temperature operating conditions of the spindles, but under these conditions its predictions are not useful, as the supervision of operating temperatures is done properly by the machine's cooling system. In contrast, this method has proved to be sufficiently accurate at identifying atypical temperature behaviors, which can permit early correction and, more interestingly, the programming of new machining conditions to avoid temperature malfunctions, and thereby, ensure a longer average life of the machine tool and its most critical elements that are monitored.

A common feature to all six problems is that competitive results were not produced by the methods that entailed higher computational costs due to the tuning of some of their parameters (i.e., SVMs and neural networks). Therefore, there is an easy and computationally low-cost way to leverage the extracted data.

The results suggest that the ensembles might be more suitable, as they predict from a voting scheme of their base classifiers. These base classifiers differ between each other, as they are trained by resampling the data set, and/or by giving more weight to some instances and input features than

others. In this way, they do not work with a single version of the data set, but with as many as there are base predictors (100 in the experiments), which will somehow generalize patterns that might not have been identified.

Finally, three future works are proposed. The first one will be the exploration of multi-label methods [57] to try to improve the results for the mode classification and the four regression problems. Secondly, the identification of the programmer's footprint will be on the research focus; that is, the subtle characteristics of each machine programmer that can optimize machining programs. Thirdly, the implementation of these methods will be in a cloud service connected to the IoT FIoT platform, as described in this paper. An innovative development that will integrate the entire process on one system: from data acquisition to the extraction of useful information for the end user.

**Author Contributions:** conceptualization, D.P.-G. and D.A.-D.; methodology, D.P.-G. and J.M.; software, J.L.G.-L. and D.P.-G.; validation, J.L.G.-L. and J.M.R.-S.; formal analysis, J.M.R.-S. and J.M.; investigation, J.L.G.-L. and J.M.R.-S.; resources, D.P.-G. and D.A.-D.; data curation, D.P.-G. and D.A.-D.; writing—original draft preparation, J.L.G.-L. and J.M.R.-S.; writing—review and editing, J.L.G.-L. and J.M.R.-S.; visualization, J.L.G.-L. and J.M.; supervision, D.P.-G. and D.A.-D.; project administration, J.M.R.-S. and D.A.-D.; funding acquisition, D.A.-D. and J.M. All authors have read and agreed to the published version of the manuscript.

**Funding:** This investigation was partially supported by the Projects TIN2015-67534-P (MINECO/FEDER, UE) of the Ministerio de Economía Competitividad of the Spanish Government and projects CCTT1/17/BU/0003 and BU085P17 (JCyL/FEDER, UE) of the Junta de Castilla y León, all of them co-financed through European-Union FEDER funds.

**Acknowledgments:** Our thanks to the company Mipromec S.C. for all the help with capturing data from their manufacturing facilities.

**Conflicts of Interest:** The authors declare no conflict of interest.

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
