# Peer review of "Using Ensembles for Accurate Modelling of Manufacturing Processes in an IoT Data-Acquisition Solution"

_applsci, doi:10.3390/app10134606_

Round 1

Reviewer 1 Report

The authors have widely improved the paper and I believe that it is ready to be published.

Author Response

Thank you very much

Reviewer 2 Report

The paper proposed a prediction model using the machine learning techniques based on the information extracted from an IoT platform connected to machining center. It also evaluated which algorithmic technique might be the best to build accurate prediction models. The strategy is validated for the classification of the state of a machining center, its working mode, and the prediction of the thermal evolution of the axis motors and the milling head motor.

The paper is interesting and well organized, and the experimental study is thorough. However, the methodology of comparing different machine learning algorithms should be further validated. I’m not sure it is a fair comparison to set each algorithm with only one configuration, since changing the hyperparameters of the machine learning algorithms may change the comparison results. An alternative way is to setup criteria for good enough prediction (based on industrial standard), and then test the machine learning algorithms under what kind of (or there is no) configurations can provide good enough prediction.

Reviewer 3 Report

Authors studied the usage of different machine-learning techniques for information extraction from a machine connected IoT platform by experiments. These machines are working under realistic industrial conditions. They aim to evaluate which algorithmic technique might be the best to build accurate prediction models for the optimization of machining processes. As a result, they found out that Rotation Forest-based ensembles have the best performance in the experiments for all the metrics under study.

The paper is good, and the technical details are provided sufficiently to support the arguments. Experiment results are also informative and it makes the paper contribution sufficient to the current state of the art. However, Paper presentation is needed to be improved by reorganizing some sections and removing the repeated sentences. In general, I found the paper and the results very interesting and innovative. Therefore, I recommend it for the publication after addressing the revision according to the comments below:

(1) In the manuscript, a lot of text is highlighted green. Please correct it.
(2) The introduction is hard to follow as in every subsection, there is a lot of related work written in between. So, write the related work as a different section or subsection whatever is suitable.
(3) The conclusion should be concise and highlight only the main point not the story and reason behind it. In the paper, it is very detailed that it can confuse the reader about paper contribution. Moreover, some information can be skipped as they are repeated or transferred into the introduction section. I would suggest using bullet points to write the main contribution and if needed then also add the explanations.
(4) spaces between some lines are quite close e.g., on pages 9, 10. Make a proper space and check this throughout the paper.

Round 2

Reviewer 3 Report

I am glad to see that the authors revised the manuscript with proper care and attention. Now the paper is well organized and in a publishable format. Therefore, I recommend the paper to publish in its current form.

This manuscript is a resubmission of an earlier submission. The following is a list of the peer review reports and author responses from that submission.

Round 1

Reviewer 1 Report

The present manuscript describes a very nice application of currently existing techniques to the development of an IoT platform.

I see no scientific novelty in this work, other than a very nice application to an industrial problem. However, no advancement in the state of the art is found.

In view of this, I regret to ask the editor not to accept the work in its present form.

Reviewer 2 Report

In this paper, the authors investigate the new machine learning algorithm to extract data in the IoT platforms. However, it does not show the novelty of their work since section 2 and 3 are from other works that are about the IoT platforms. From my perspective, they just simply implement existing method to classify data that achieved from IoT platforms. The presentation of the paper is not well written, there is no summary of contributions, or what they have done in this work. 

Reviewer 3 Report

In this study, the authors compare different machine learning techniques to extract information from a machining center. In addition, they use real industrial data from a workshop which make this study very interesting for future real purposes. The study design is appropriate because the authors compare several machine learning techniques to test which ones work best. I think this study is interesting and within the scope of the journal, although some minor revisions should be made before it is accepted.

Major comments

English must be checked carefully. In general, there are many mistakes with plurals and singulars and also with the past tenses (e.g. Line 57). In the specific comments there are some mistakes that I have noticed but there are more. Please, check the English of the entire manuscript.

Introduction (from lines 50 to 73). I find the discussion difficult to follow. First, the authors introduce the need of machine learning techniques, then talk about specific examples and finally again about the importance of machine learning techniques. These paragraphs should be rewritten to provide a clear introduction to their work.

In general, I believe that the first paragraph of Section 4.1 is difficult for the reader to follow. The information of Table 3 is presented (which should be also explained in the Table’s caption) but still is unclear. The authors also talk about two ranking problems in Section 4.1 which are not mentioned anywhere else and the same goes for wins and losses. Please check this paragraph and try to be more explicit.

The captions of the tables should be improved. Specifically, In Tables 3, 4 and 5 there are two columns with “state” and “mode” which do not correspond with the true values of “state” and “mode” (the states go from 1 to 4 and the modes from 1 to 5). In this case I would either be more explicit in the caption or write F-macro for state and for mode instead of just state and mode.

I find the explanation related to the temperatures confusing. For example, in line 463 the authors state that “the temperatures are not deviated by more than 1.5 degrees from the temperature of the median” which is confusing because that statement is in general true for the 1st and 3rd percentile but not for all temperatures since the whiskers are more than 5 degrees of difference compared to the median value.

Specific comments

Lines 28 and 29. Before using for the first time the acronyms PLC and CNC, they should be introduced in the text.

Line 36: Please, correct the mistake with the English and plurals in several lines: Line 36 - “have opened”; Line 47 – refer to.

Line 48: not which but whose or where

Line 37: The word “and” appears twice.

Line 51: “Umbalanced” or “imbalanced”?

Line 99: If you name a library or python there should be a reference to this.

Line 106: Why is “as” written in uppercase? Also in line 120.

Line 106: It is a bit confusing the purpose of Gateway IIoT. Please, clarify this.

Figure 2: First, there is no label for the Y axis. I know it is temperature but this should appear.

Reviewer 4 Report

The presented research explores a large variety of techniques that could be applied to IoT manufacture in order to retrieve information with machine learning.

The same results, as it is said, could be difficult to achieve by adding new sensors and actuators to the systems, since the industrial machinery is often realized in environments in which it is very hard to apply modifications.

I don't have particular methodological failures to highlight, since the paper appears to be well structured, with a relevant theoretical part followed then by the tests and the evaluation part that is also explained.

I would also suggest a revision of the English language.

My minor points:

ABSTRACT: suddenly, at line 12, you introduce the temperature of four motors. This is not clear at all, and the reader is very likely not understanding what you are talking about.

INTRODUCTION: line 21, a typo: have -> has (or, alternatively, the sentence is to be rephrased).

Line 57, "have proven their ability"

Line 77: missing explanation of what is the 'F' in FIoT

DATA ACQUISITION AND SETUP: Line 92, would you spend some lines more in explaining your setup? "five axis machining center  equipped with..." may be better described, since not all readers are expert of this instrumentation.

Line 99: please, a reference to the web to the pyjh library would be handy here

MODELING: In table 1, you said that output variables are outlined in bold, but actually this is not the case (or, alternatively, it's very hard to see): please check.

Line 209: rephrase here, it's not clear

Line 223: please rephrase here

Reviewer 5 Report

The article presents a comprehensive comparison of various machine learning algorithms for processing data from a small-scale workshop use-case. Indeed, industrial data processing is not exhaustively explored and the depricated control technology such as PLC do not facilitate the data collection and processing. The pros and cons of each algorithm are adequately described and the results are well presented and support the higher performance of the ensemble-based methods.

1) I am confused what the novelty and the contribution of the article is. A simple workshop setup is used, which includes a machining center and the volume of generated data is not huge (a 3-month set is used). On the other hand, the authors  do not propose a new or modified machine learning approach for classification and prediction, but they compare already proposed algorithms. The  authors should clarify if their article has significant theoretical contribution or proposed a novel IoT solution for industrial setting.

2) The introduction is vague regarding the contributions of the article. The authors should clearly present the objectives and the contribution of the article.

3) There are many references in the Introduction, which are briefly presented or just cited. The authors should add an individual section about related work and present the most relative studies on industial IoT platforms for data collection and processing and machine learning (or not)  techniques on processing unbalanced data sets. Also they should explicitly mention which is the novelty of the article with respected to the above studies.

4) Line 404 and 494. The term "Status" is used instead of "State". Is this correct?

5) Lines 406-407. The authors wrote "When these values are followed by “*” they are significantly worse than the one achieved by the first ranked method." In Table 3, the first ranked method has the same score with Bagging of Random Balance of C4.5 with *. Why is it better? Alsoi the Bagging of Random Balance of C4.5 has higher rank for Mode. I think the definition and usage of * is confusing and must change.

6) I advise the authors to replace the term "issues: with "remarks or conclusions" in section 4.